# Syntheses of 25-Adamantyl-25-alkyl-2-methylidene-1α,25-dihydroxyvitamin D_3_ Derivatives with Structure–Function Studies of Antagonistic and Agonistic Active Vitamin D Analogs

**DOI:** 10.3390/biom13071082

**Published:** 2023-07-06

**Authors:** Kazuki Maekawa, Michiyasu Ishizawa, Takashi Ikawa, Hironao Sajiki, Taro Matsumoto, Hiroaki Tokiwa, Makoto Makishima, Sachiko Yamada

**Affiliations:** 1Department of Chemistry, Faculty of Science, Rikkyo University, Toshima-ku, Tokyo 171-8501, Japan; kkazukki@gmail.com; 2Division of Biochemistry, Department of Biomedical Sciences, Nihon University School of Medicine, Itabashi-ku, Tokyo 173-8610, Japan; ishizawa.michiyasu@nihon-u.ac.jp; 3Laboratory of Organic Chemistry, Gifu Pharmaceutical University 1-25-4 Daigaku-Nishi, Gifu 501-1196, Japan; ikawa-ta@gifu-pu.ac.jp (T.I.); sajiki@gifu-pu.ac.jp (H.S.); 4Department of Functional Morphology, Division of Cell Regeneration and Transplantation, Nihon University School of Medicine, Tokyo 173-8610, Japan; matsumoto.taro@nihon-u.ac.jp

**Keywords:** vitamin D, nuclear receptor, osteoblast differentiation, cell selectivity

## Abstract

The active form of vitamin D_3_, 1α,25-dihydroxyvitamin D_3_ [1,25(OH)_2_D_3_], is a major regulator of calcium homeostasis through activation of the vitamin D receptor (VDR). We have previously synthesized vitamin D derivatives with large adamantane (AD) rings at position 24, 25, or 26 of the side chain to study VDR agonist and/or antagonist properties. One of them—ADTK1, with an AD ring and 23,24-triple bond—shows a high VDR affinity and cell-selective VDR activity. In this study, we synthesized novel vitamin D derivatives (ADKM1-6) with an alkyl group substituted at position 25 of ADTK1 to develop more cell-selective VDR ligands. ADKM2, ADKM4, and ADKM6 had VDR transcriptional activity comparable to 1,25(OH)_2_D_3_ and ADTK1, although their VDR affinities were weaker. Interestingly, ADKM2 has selective VDR activity in kidney- and skin-derived cells—a unique phenotype that differs from ADTK1. Furthermore, ADKM2, ADKM4, and ADKM6 induced osteoblast differentiation in human dedifferentiated fat cells more effectively than ADTK1. The development of vitamin D derivatives with bulky modifications such as AD at position 24, 25, or 26 of the side chain is useful for increased stability and tissue selectivity in VDR-targeting therapy.

## 1. Introduction

Osteoporosis is a disease in which the density and quality of bone are reduced [1]. The loss of bone occurs silently and progressively, and often there are no symptoms until the first fracture occurs. Worldwide, osteoporosis causes more than 8.9 million fractures annually, and 1 in 3 women and 1 in 5 men over the age of 50 will experience osteoporosis.

Muscle cells, such as cardiac cells, use calcium ions (Ca^++^) as a messenger for contraction. The extracellular concentration of Ca^++^ (1 mM) is tightly regulated and kept as high as 10^4^ times that of their intracellular counterpart [2,3]. This concentration difference is important for the proper function of Ca^++^. Bone is a dynamic storehouse of calcium, and the active form of vitamin D_3_—1α,25-dihydroxyvitamin D_3_ [1,25(OH)_2_D_3_] [4,5]—and parathyroid hormone (PTH) [6] are major regulators of calcium homeostasis. Vitamin D_3_ analogs, such as 1α-hydroxyvitamin D_3_ [7] and eldecalciferol [8,9,10], have been used as good therapeutic agents for osteoporosis. For a long time now, the usefulness of the active vitamin D_3_ and its derivatives in the treatment of osteoporosis has not been accepted in the USA and European countries because of the potential adverse effect of hypercalcemia. In these countries, bisphosphonates [11] are the agents most commonly used for osteoporosis. Bisphosphonates are pyrophosphate analogs and bind strongly to hydroxyapatite in bone. Because of their non-hydrolysable P-C-P structure, bisphosphonates accumulate in osteoclasts and can cause significant side effects, such as osteonecrosis of the jaw [12]. Other drugs used for osteoporosis include peptides of the parathyroid hormone family [13], as well as the antibody drug denosumab [14]. Recently, many reports and reviews have recommended the use of vitamin D combined with Ca for the prevention of osteoporosis and bone fractures [15,16]. Bislev et al. [17] reported that vitamin D_3_ supplementation (70 μg [2800 IU]/day) improves strength and trabecular thickness in the tibia, as well as volumetric bone mineral density in the trochanter and femoral neck, compared with those given placebo, but does not affect aerialbone mineral density. They also reported increases in the plasma concentrations of both 25-hydroxyvitamin D_3_ [25(OH)D_3_] and 1,25(OH)_2_D_3_ in women treated with vitamin D_3_. Vitamin D_3_ is produced from 7-dehydrocholesterol (provitamin D_3_) in the skin via sunlight irradiation [18], followed by thermal isomerization, and converted to 1,25(OH)_2_D_3_ via metabolic hydroxylations at position 25 [19], followed by the 1α-position [4,20]. However, the production of vitamin D_3_ in the skin [21] and its metabolism to the active form 1,25(OH)_2_D_3_ decrease age-dependently [22]. Therefore, supplementation of not only vitamin D_3_ but also the active metabolite 1,25(OH)_2_D_3_ should be necessary for the prevention of age-related osteoporosis and bone fractures. More broadly, vitamin D has recently been recognized as an important agent in controlling not only bone density [22], but also the rate of aging and age-related diseases [23].

Vitamin D derivatives with selective modulatory activity on the vitamin D receptor (VDR) [24,25,26] have been developed for the treatment of VDR-related diseases, including bone and calcium diseases, malignancies, immune and inflammatory diseases, and metabolic disease. We have been developing active analogs of vitamin D from the viewpoint of synthetic chemists. We reported for the first time (1) the synthesis of super-high radioactive [^3^H]-25-OHD_3_ and its biological conversion to [^3^H]-1,25(OH)_2_D_3_ to prove the presence of the VDR in various target tissues [27,28,29]; (2) the synthesis of 24F_2_-25(OH)D_3_ and 24F_2_-1,25(OH)_2_D_3_ [30,31] to examine the importance of 24-hydroxylation in the vitamin D_3_ metabolism; (3) the activity of 24F_2_-1,25(OH)_2_D_3_ with four to seven times more activity than natural metabolite (**1**) in HL-60 cell differentiation [32]; and (4) the synthesis of the highly fluorescent dienophile to selectively bind to the s-*cis* 5,10(19)-diene structure of vitamin D and quantify vitamin D_3_ metabolites in biological fluids [33,34,35]. We have been interested in the role of the VDR residues lining the ligand-binding pocket, as understanding this would allow us to clarify which residues are important for ligands to express the VDR activity. To this end, we prepared VDR one-point alanine mutants for all 34 residues lining the ligand-binding pocket (LBP) of the VDR, and the effect of the mutation was evaluated in transactivation assays using an hVDR expression vector and a luciferase reporter gene with the mouse osteopontin VDRE at the promoter in COS7 cells [36,37]. From these studies, we showed which LBP residues are important in expressing the activity of the ligands.

On the basis of the studies described above, we recently synthesized vitamin D derivatives ADXY (XY: variable) substituted with a large adamantane (AD) ring at position 24, 25, or 26 of the side chain to develop VDR ligands with antagonist and/or partial agonist properties [38,39,40,41,42,43,44] ([Fig biomolecules-13-01082-ch001]). We designed the ADXY compounds based on the idea that a large adamantane ring facing the VDR activation function 2 part (human helix 11 to helix 12) would prevent this part from keeping hold of the active conformation. In addition, we thought that a double or a triple bond at the side chain would be necessary to fix the side chain conformation more tightly. These ADXY compounds with a 22-double bond have been found to act as antagonists of 1,25(OH)_2_D_3_ (**1**): ADTT **3b**, VDR affinity 67% of that of **1**, transcriptional activity EC_50_ 10^−9^ M, efficacy 14% of that of **1** [40,41]; ADMI3 **4b**, VDR affinity 17% of that of **1**, transcriptional activity, EC_50_ 2 × 10^−8^ M, efficacy 10% of that of **1**, antagonist activity for **1** IC_50_ 3 × 10^−9^ M [38] (Table 1). On the other hand, analogs with 23,24-triple bond ADTK1 (5b) (VDR affinity 90% of that of **1**, transcriptional activity EC_50_ 10^−9^ M, efficacy 81% of that of **1**) [42], and ADOR1 (**6b**), ADOR2 (**6a**) [43], and ADYW2 (**7a**) [44], also showed partial agonist activities: ADTK1 (**5b**), VDR affinity 90% of that of 1, transcriptional activity EC50 10^−9^ M, efficacy 81% of that of 1 [42] (see also: Table 1).

In this paper, we report the synthesis and biological activity of analogs of ADTK1 (**5b**) with a methyl, ethyl, or n-butyl group (**8a**, **8b**, **9a**, **9b**, **10a** and **10b**) at position 25. We expect these compounds to show more selective activities because of the inserted hydrophobic alkyl group. Elevation of hydrophobicity at the side chain of the vitamin D compounds is known to improve their activity. We also investigated the structure–activity relationships of all ADXY compounds on the VDR antagonist and agonist relationships, comparing them with the known antagonists TE-9647 and ZK168281 and agonists KH1060 (**15**) and 24-carboranyl-1α-(OH)D_3_ analog (**16**) via an analysis of their crystal structure ([Fig biomolecules-13-01082-ch002]). We also carried out computational analyses using the fragment molecular orbital (FMO) computational method and inter-fragment interaction energy (IFIE) as a tool.

## 2. Results and Discussion

### 2.1. Synthesis of (25R)- and (25S)-25-adamantyl-25-alkyl-2-methylidene-23-yne-19-norvitamin D Compounds (***8a***, ***8b***, ***9a***, ***9b***, ***10a*** and ***10b***)

We synthesized novel vitamin D derivatives (**8a**, **8b**, **9a**, **9b**, **10a** and **10b**) from 25-ketone **11** [42] via alkylation with R’Li (R’ = methyl, ethyl, and n-butyl), followed by deprotection of the 1- and 3-hydroxyl groups under acidic conditions (camphorsulfonic acid in methanol; Figure 1). The alkylation yielded a mixture of the epimers at C(25) at an approximately 1:1 ratio; the epimers were then separated via HPLC (YMC-Pack ODS-AM). The stereochemistry at C-25 of each pair of 25-alkylated analogs (**8a**, **8b**, **9a**, **9b**, **10a** and **10b**) was tentatively assigned on the basis of their biological activities in comparison with the non-alkylated analogs ADTK1 (**5b**) and ADTK2 (**5a**). The affinity for human VDR of each 25-epimer pair (**8a** and **8b**, **9a** and **9b**, and **10a** and **10b**) differed greatly. For example, one group of the epimers was eluted more rapidly (**8a**, **9a**, and **10a**) than the others (**8b**, **9b**, and **10b**) via HPLC (reversed-phase column) and showed much lower activities (5–11-fold) than the others, tentatively assigned as 25*R* epimers—the same 25*R*-configuration as the non-alkylated analog ADTK2 (**5a**) with a 25*R*-configuration; **5a** (25*R*) was 24 times less potent than **5b** (25*S*). The latter isomers (**8b**, **9b**, and **10b**) were suggested to have the same 25*S*-configuration as that of the more active 25*S*-isomer ADTK1 (**5b**).

### 2.2. VDR-Binding Affinity

The VDR-binding affinities of 25-alkylvitamin D analogs ADKM1-6 (**8a**, **8b**, **9a**, **9b**, **10a** and **10b**) were evaluated using recombinant hVDR-LBD via competitive binding between [^3^H]-1,25(OH)_2_D_3_ and analogs (Figure 1). ADKM2, 25*S*(**8b**) showed the highest affinity (IC_50_ 0.6 nM), 67% that of 1,25(OH)_2_D_3_ (IC_50_ 0.4 nM). ADKM4 (**9b**) and ADKM6 (**10b**) analogs exhibited the second- (IC_50_ 1.3 nM, 27%) and the third-highest affinities (IC_50_ 1.6 nM, 23%), respectively, and the other three analogs with a 23*R*-configuration—ADKM1 (**8a**), ADKM3 (**9a**), and ADKM5 (**10a**)—exhibited much lower affinities (IC_50_ 6.2 nM, 6%; IC_50_ 6.7 nM, 6%; and IC_50_ 23.4 nM, 2%, respectively).

### 2.3. VDR Transactivation Activity

The transcriptional activity of the 25-alkylvitamin D analogs on the VDR was evaluated via a luciferase reporter assay in HEK293 human kidney cells transfected with an hVDR expression vector and a luciferase reporter with mouse osteopontin vitamin D response elements (VDREs, SPP×3-tk-LUC). Analogs ADKM2 (**8b**) and ADKM4 (**9b**) showed the highest activity (EC_50_ 0.2 nM and 0.2 nM, efficacy 101% and 104%, respectively) among the six analogs and similar activities to that of 1,25(OH)_2_D_3_ (0.2 nM, efficacy 100%) (Figure 2). ADKM1 (**8a**) and ADKM6 (**10b**) exhibited the next-highest activity (0.53 nM and 0.62 nM, efficacy 78% and 95%, respectively). The EC_50_s for ADKM3 (**9a**) and ADKM5 (**10a**) were 2.33 and 2.63 nM, respectively, and their efficacies were 90% and 83%, respectively. The results showed that all of the compounds had agonistic properties.

### 2.4. Effects on Interactions of the VDR with RXRα and Cofactors

The effects of 25-alkylvitamin D analogs on the VDR when interacting with the heterodimer partner retinoid X receptor α (RXRα), steroid receptor coactivator 1 (SRC-1), and nuclear receptor corepressor 1 (NCoR) proteins were evaluated using mammalian two-hybrid assays, and the results were compared with the effect of 1,25(OH)_2_D_3_ (1). In the interaction of the VDR with RXRα, ADKM2 (**8b**) showed the highest potency (EC_50_ 0.1 nM), which was similar to that of the natural hormone **1** (0.1 nM) (Figure 3A). ADKM6 (10b) showed the second-highest activity, and ADKM1 (**8a**), ADKM3 (**9a**), ADKM4 (**9b**), and ADKM5 (**10a**) showed EC_50_ values of 1.1, 1.1, 1.1, and 5.2 nM, respectively. The efficacy was highest for ADKM6 (**10b**), at 114% compared to natural hormone **1**, and values of 59, 59, 51, 56, and 51% were obtained for the other compounds, respectively. In the VDR’s binding to SRC-1, ADKM2 (**8b**) showed the highest activity (EC_50_ 0.6 nM) among the six compounds, similar to that of natural hormone **1** (0.5 nM), while ADKM4 (**9b**) and ADKM6 (**10b**) showed the second- and third-highest activities (1.2 and 2.4 nM), respectively (Figure 3B). The potencies of the other compounds—ADKM1 (**8a**), ADKM3 (**9a**), and ADKM5 (**10a**)—were 8.4, 10.2, and 2.4 nM, respectively. In the dissociation of NCoR from the VDR, similar to the recruitment activity with SRC1, the analog ADKM2 (**8b**) showed the highest activity (IC_50_ 0.4 nM), followed by ADKM6 (**10b**) and ADKM4 (**9b**) (0.4 and 0.5 nM, respectively), when compared to natural hormone **1** (0.3 nM) (Figure 3C). The other compounds—ADKM1 (**8a**), ADKM3 (**9a**), and ADKM5 (**10a**)—showed lower activities (2.3, 6.3, and 8.5 nM, respectively). These results for transcriptional activity and cofactor interaction are consistent with their affinities for the VDR. ADKM2 (**8b**) had a lower affinity for the VDR than compound **1**, but it exhibited similar levels of VDR transcriptional activity, suggesting that its intracellular stability or interaction with other cofactors is superior to that of **1**.

### 2.5. Effects on Endogenous Gene Expression in Various Tissue Cells

To investigate the tissue-selective effects of ADKM1-6 (**8a**, **8b**, **9a**, **9b**, **10a** and **10b**), we examined the expression of the VDR target gene *CYP24A1*, which encodes cytochrome P 450 24A1 (CYP24A1), in various cells: kidney-epithelium-derived HEK293 cells (Figure 4A), intestinal-mucosa-derived SW480 cells (Figure 4B), myeloid-derived U937 cells (Figure 4C), skin-keratinocyte-derived HaCaT cells (Figure 4D), and osteoblast-derived MG63 cells (Figure 4E). In addition, we examined *CYP24A1* expression in lung mucoepidermoid carcinoma-derived H292 cells (Figure 4F). We compared the effects of the compounds (**1**, **8a**, **8b**, **9a**, **9b**, **10a** and **10b**) at 100 nM on *CYP24A1* expression to that of the natural hormone 1. In a gene expression analysis, ADKM2 (**8b**) showed the most diverse tissue selectivity among the test compounds. ADKM2 (**8b**) induced *CYP24A1* expression more strongly in U937 blood cells (77.8%, relative to natural hormone 1 100%), HaCaT skin cells (85.6%), and MG63 bone cells (63%) than in other tissues, such as HEK293 kidney cells (33.2%), SW480 intestine cells (42.3%), and H292 lung cells (50.1%). ADKM4 (**9b**) and ADKM6 (**10b**) induced *CYP24A1* expression as much as or more than natural hormone **1** (100%) in all cells: HEK293 kidney cells (91.5% and 61.4%, respectively), SW480 intestinal cells (126% and 72.9%, respectively), U937 blood cells (85.1% and 77.2%, respectively), HaCaT skin cells (91.9% and 72.5%, respectively), MG63 bone cells (73.3% and 54.2%, respectively), and H292 lung cells (113% and 87.9%, respectively). ADKM1 (**8a**), ADKM3 (**9a**), and ADKM5 (**10a**) were weakly effective in all cells, with 10-50% activity. We previously reported that AD47 (**2**) with an adamantyl group at position 25 interacts with the coactivator DRIP205 as much as compound **1** and interacts less with SRC1 [39]. AD47 (**2**) selectively induced *CYP24A1* expression in HaCaT cells (50% compared to **1**) and intestine-derived HCT116 cells (75%), which highly express DRIP205. The cell-selective (or tissue-selective) action of vitamin D analogs may be influenced by the expression pattern of cofactors—a mechanism similar to the tissue-selective ER action of tamoxifen in the mammary glands and uterus [49]. In contrast, we previously reported the non-alkylated vitamin D derivative ADTK1 (**5b**), like compound **1**, strongly interacts with SRC1 and induces *CYP24A1* expression most strongly in HEK293 cells, which express high levels of SRC1, but less than 50% in HaCaT cells, which express less SRC1 [42]. The cell-selective VDR activity of ADTK1 (**5b**) is suggested to be determined by the expression of cofactors, such as SRC1, in each cell. In this study, despite the compound ADKM2 (**8b**) interacting with SRC1 as strongly as natural hormone 1 and ADTK1 (**5b**), the induction of *CYP24A1* expression in HEK293 cells was weaker than in HaCaT cells. The cell-selective VDR activation mechanism of ADKM2 (**8b**) may be affected by intracellular stability and/or interaction with coactivators other than SRC1. The analogs ADKM4 (**9b**, 27%) and ADKM6 (**10b**, 23%), with elongated alkyl groups at C-25, had weaker VDR affinities than ADKM2 (**8b**, 67%) but similar VDR transcriptional activation efficacies (104% and 95%, respectively), and their ability to induce CYP24A1 expression in all cell lines was comparable to that of natural hormone **1**. These results indicate that the elongation of the 25-alkyl group masks the cell selectivity of ADKM2 (**8b**). ADKM4 (**9b**) and ADKM6 (**10b**), which have weaker affinities for the VDR, may be more stable in cells than natural hormone **1** or ADKM2 (**8b**).

### 2.6. Osteogenic Differentiation Activity in Human Dedifferentiated Fat Cells

Mature adipocytes from the adipose tissue of mammals such as humans and rodents can be dedifferentiated into pluripotent cells using ceiling culture [50]. Dedifferentiated fat (DFAT) cells have potential applications in both regenerative medicine and injury healing, as well as mesenchymal stem cells [51,52], embryonic stem cells, and induced pluripotent stem cells [53]. The osteogenic activity of vitamin D derivatives was assessed based on alkaline phosphatase (ALP) activity. In DFAT cells, the addition of natural hormone **1** to the osteogenic medium induced ALP activity (100%) (Figure 5). Among the derivatives, ADKM2 (**8b**) was the strongest in inducing ALP activity (98.8%), with ADKM4 (**9b**) and ADKM6 (**10b**) showing the next-strongest effects (72.6% and 62.4%, respectively). The other compounds (ADKM1 (**8a**), ADKM3 (**9a**), and ADKM5 (**10a**)) also showed weak activity of less than 50% (41%, 40.4%, and 19%, respectively). ADKM2 (**8b**), ADKM4 (**9b**), and ADKM6 (**10b**) showed stronger activities in inducing bone differentiation than ADTK1 (**5b**) (45.6%). The results show that alkylated vitamin D analogs with VDR affinity less than that of natural hormone **1** can induce osteoblast differentiation. We previously reported that the vitamin D derivative ADYW2 is more stable than natural hormone **1** in bone-derived MG63 cells [44]. The introduction of adamantyl groups may affect the stability of vitamin D analogs.

### 2.7. Structure–Activity Relationship of VDR Agonists and Antagonists

We synthesized 25-epimeric isomer pairs of 25-methyl- (ADKM1 **8a** and ADKM2 **8b**), 25-ethyl-(ADKM3 **9a** and ADKM4 **9b**), and 25-n-butyl-25-adamantylvitamin D analogs (ADKM5 **10a**, ADKM6 **10b**) of the ADTK1 (**5b**) [42] VDR partial agonist. The stereochemistry at C-25 of the higher-VDR-binding isomers (**8b**, **9b**, and **10b**) was determined as 25S, and that of the lower-activity isomers (**8a**, **9a**, and **10a**) as 25*R*, as described in the Results section.

Since we do not have crystal structures of ADKM1-6 (**8a**, **8b**, **9a**, **9b**, **10a** and **10b**)/VDR complexes, we constructed the three-dimensional structures of those rVDR complexes via a computational method using the X-ray crystal structure data (3vtb) of ADTK1 (**5b**) [42] as a model (see the Materials and Methods section). The calculated model structures of the three compounds ADKM2, ADKM4, and ADKM6 (**8b**, **9b**, and **10b**, respectively) complexed with rat VDR are shown in Figure 6. In all of the computational structures, the two His residues 301 and 393 are placed within hydrogen-bonding distance with respect to the 25-hydroxy group. The three Leu residues (Leu223, Leu400, and Leu410) and the adamantane ring of the ligand are situated in the agonistic positions, interacting with one another similarly to the rVDR/1,25(OH)_2_D_3_ (**1**) complex (Figure 7A). The 25-methyl and ethyl groups were inserted into a pocket of the VDR where they did not interfere with nearby residues. Only the n-butyl group of ADKM6 (**10b**) interfered with Phe418, which then would change the conformation to adopt the n-butyl group in the pocket, as shown in Figure 6D. The introduction of methyl, ethyl, and n-butyl groups at C-25 increases the hydrophobic interaction energy (FMO, inter-fragment interaction energies (IFIEs)) of Phe418 in ADTK2 (**8b**), ADKM4 (**9b**), and ADKM6 (**10b**) to −6.34, −7.40, and −10.95 kcal/mol, respectively, when given −3.64 kcal/mol of ADTK1 (**5b**) via the FMO IFIE calculation (Appendix A). ADKM6 (**10b**) showed the highest activity in an assay to examine the effects of the ligands on the VDR to bind to RXR. The 25*S*-ethyl substitution had the highest effect on the *CYP24A1* induction in kidney (HEK293), intestine (SW480), and lung (H292) cell lines.

We postulate that three Leu residues—Leu223, Leu400, and Leu410 (rVDR)—are the key residues for the VDR to form the active conformation and direct a VDR ligand to act as an agonist or antagonist. We reported this motif [41], without recognizing its importance, back in 2008 in an X-ray crystallographic analysis of new-type vitamin D antagonist compounds 25- or 26-adamantyl-22,23-didehydro1α,25-dihydroxyvitamin D analogs ADTT (**3b**) [40,41] and ADMI3 (**4b**) [38]. However, we later found that ADVD analogs have a variety of biological properties, ranging from antagonist to agonist activities (Table 1). We reported that the 25-adamantyl-1α,25-dihydroxy-23,23,24,24-tetradehydro-VD analog ADTK1 **5b** [42] showed partial agonistic activity, with VDR affinity IC_50_ 0.5 × 10^−9^ M and transcriptional activity EC_50_ 1 × 10^−10^ M (69% efficacy)—a higher potency than that of the natural hormone (**1**). In the present study, we examined and suggested a structure–activity relationship with active vitamin D analogs, focusing on agonist and antagonist activities. We analyzed antagonistic and agonistic VD compounds complexed with VDR, whose structures were unequivocally determined via crystallographic analysis, such as with the antagonists ADTT (**3b**), TEI-9647 (**15**) [45,54], and ZK168281 (**16**) [46,47] and the agonists ADTK1 (**5b**), KH1060 (**17**) [55], and carborane VD (**18**) [48], as shown in Table 1. In addition, we analyzed the physicochemical properties of VDR complexes of adamantane VD compounds with agonistic and antagonistic properties, using a theoretical ab initio FMO computational method and FMO IFIE as a tool.

In the 1,25(OH)_2_D_3_/rVDR complex, we first chose 18 important residues: 10 from the residues interacting with the side-chain part of the ligand (Phe418, Val414, Leu410, Leu400, Tyr397, His393, His301, Ala399, Ile260, and Leu223), 4 from helix 12 (Val417, Glu416, Leu415, and Thr411), 1 from helix 5 (Ile264), and 3 from helix 3 (Val230, Ser231, and Ile234) (Figure 7B and Appendix A). These residues and ligands interact with one another to form the active VDR conformation (Figure 7B). In the VDR, the A-ring of 1,25(OH)_2_D_3_ (**1**) is placed in the innermost part of the ligand-binding pocket (LBP), where the two 1α-and 3β-hydroxy groups form four hydrogen bonds, each with two residues: Ser233 and Arg270, and Tyr143 and Ser273, respectively (Figure 7C). In addition, we clarified the reason why the exocyclic methylene group at C-2 of the adamantane vitamin D analog enhanced its activity significantly (50 times that of the VDR affinity in ADNY93 (**3a**) compared with AD47 (**2**)) (Table 1). The methylene proton forms H–π bonds (3.98–5.34 Å) with the aromatic ring of Tyr143 (rVDR), which forms hydrogen bonds with the 3β-hydroxyl group, as shown in Figure 7C.

When a ligand binds with a VDR (rat), Leu400 at the N-terminal part of helix 11 and Leu410 at loops 11–12 link to the C-26 of the ligand; then, loops 11–12 bend, and both Leu residues bind to Leu227 at the N-terminal of helix 3, thereby closing up the VDR to form the active conformation (Figure 7A).

The 23,23,24,24-tetradehydro-VD analog ADTK1 (**5b**) showed partial agonistic activity, in contrast to the 22,23-didehydro analog ADTT (**3b**) (Table 1), which had antagonistic properties. ADTK1 (**5b**)/rVDR is shown in Figure 7D at the position of the three Leu’s. Here, Leu223 and Leu400 strongly interact at their Cδ, with Å distances of 3.985, 4.481, and 4.195; in addition, Leu223, Leu400, and a secondary AD carbon of the ligand interact at Å distances of 3.985, 3.525, and 3.493, respectively, forming a rigid triangular relationship. Figure 7F shows the hVDR complex of the superagonist 1,25-dihydroxy-20-epi-22-oxa-24,26,27-trihomo vitamin D analog KH1060 (**15**), overlaid with 1,25(OH)_2_D_3_ (**1**)/hVDR. The two VDR complexes are well overlapped, regardless of the bulky side chain of **15**. This shows that the bulky-but-flexible side chain can be sufficiently placed within the VDR pocket. Carboranyl VD (**16**) is a superagonistic analog with a carborane cluster at C-24 in the place of the adamantane ring of the ADVD compounds [55]. Carborane (C_2_B_10_H_12_) is a boron cluster molecule, and in the 24-carboranyl VD/zVDR complex it interacts with two His residues, His301 and His393, instead of the 25-hydroxyl group of the natural ligand. The VDR complexes of carboranyl VD(**15**)and ADTK1 (**5b**) are well overlapped, as shown in Figure 7G. The VDR affinity and transcriptional activity of carboranyl VD (**15**) are reported to be similar to those of 1,25(OH)_2_D_3_ (**1**), but its calcemic effect is smaller than that of 1,25(OH)_2_D_3_ (**1**) [48].

We next discuss VD antagonists and/or partial agonists. The 25-adamantyl-22,23-didehydro analog ADTT (**3b**) has a high VDR affinity (IC_50_ 1.3 × 10^−10^M) similar to that of 1,25(OH)_2_D_3_ (**1**), and a high transcriptional activity (EC_50_ 1 × 10^−9^ M), but with low efficacy 15%, in addition to a high antagonistic activity (IC_50_ 3 × 10^−9^ M). These activity patterns should be typical for antagonists. In the crystallographic structure of ADTT (**3b**)/rVDR, the three Leu residues do not interact with one another and are placed away from one another at a longer distance than 6 Å (Figure 7H). Each of the three Leu residues interacts with the ligand’s adamantane ring. The lack of interactions among the three Leu residues was considered to be a reason for the antagonistic activity of ADTT (**3b**). TEI-9647 (**17**) was reported as the first antagonist of the VDR. For this compound, it was suggested that the α, β-unsaturated lactone structure of the TEI compound can be a target for Cys and His residues, and the nucleophilic attack of these residues was assumed to be the reason for the antagonistic activity. However, this hypothesis has not been supported. We suggest a reason for the antagonistic activity of TEI-9647 on the reduced interactions around the three Leu residues and ligands, as shown below (Figure 7I). Firstly, the position of Leu400 is significantly changed when compared with that of 1,25(OH)_2_D_3_ (**1**) and other agonist (ADTK1) VDR complexes. The distances between δ and δ’ of Leu400 and Leu223 are longer than 6.0 Å, and those from the ligand are 6.904 and 4.227 Å, respectively; thus, the interaction energies would be considerably smaller than those of the agonists. The interaction distance between Leu410 and Leu223 is 4.248 Å, and for Leu410 and the ligand it is 4.843 Å which seems to be insufficient to obtain strong binding energy.

In Figure 7J, we show the ZK168281/zVDR complex overlaid with 1,25(OH)_2_D_3_/rVDR. Here, again, Leu430 (rVDR Leu400) changes its position significantly, but Leu255 and Leu440 do not. The interaction between each Cδ and Cδ’ of Leu255 and Leu430 are 5.408 and 4.305 Å, respectively, and these interaction energies are not expected to be strong. Other interactions take place between Leu 255 Cδ and the double bond of the 25-propenoate ethyl ester (4.090 and 4.606 Å), Leu430 and the ethyl propenoate methylene carbon (3.608 Å), and Leu400 and the C(25) cyclopropane ring (4.182 Å). Thus, the interactions are not concentrated, and the total interaction energies between three Leu residues and the ligand are not expected to be strong enough to keep the active conformation.

## 3. Conclusions

We synthesized two types of 25-adamantyl-1,25-(OH)_2_D_3_ analog: one with 22,23-didehydro-type compounds (ADTT (**3b**) [40,41], ADNY93 (**3a**) [38], and ADMI3 (**4c**)) with antagonistic activity, and the other with 23,23,24,24-tetradehydro-type compounds (ADTK1 (**5b**) [42], ADYW2 (**7a**) [44], and ADOR1 (**6b**) [43]) with agonistic activity (Table 1). In the present study, we synthesized 25-alkylated analogs of ADTK1 (**5b**) as pairs of 25-epimer methyl- (ADKM1 **8a** and ADKM2 **8b**), ethyl- (ADKM3 **9a** and ADKM4 **9b**) and n-butyl-substituted analogs (ADKM5 **10a** and ADKM6 **10b**). One of each 25-epimer pair (**8b**, **9b**, and **10b**) eluted more slowly via reverse-phase HPLC than the other (**8a**, **9a**, and **10a**) showed significantly higher activities. We tentatively assigned the isomers with higher activity as 25*S* epimers in comparison with the known epimer pairs of ADTK1 and 2 (**5b** and **5a**) [42] and ADOR1 and 2 (**6b** and **6a**) [43] (Table 1), whose stereochemistry was unequivocally determined by crystal structure analysis.

In the biological study, ADKM2 (**8b**), ADKM4 (**9b**), and ADKM6 (**10b**) showed similar transcriptional activity to 1,25(OH)_2_D_3_ (**1**) and ADTK1 (**5b**). Interestingly, ADKM2 (**8b**) showed selective VDR activity in kidney-derived and skin-derived cells—a unique phenotype opposite to that of ADTK1 (**5b**). Furthermore, these compounds induced osteoblast differentiation in human dedifferentiated fat cells more effectively than ADTK1 (**5b**).

We analyzed the crystal structures of ADTK1 (**5b**) and ADTT (**3b**) in comparison with those of KH1060 (**15**), carborane VD (**16**), TEI-9647 (**17**), and ZK168281 (**18**), and we proposed that three Leu residues (223, 400, and 410) are important for forming the VDR active conformation (Figure 7). ADTK1 (**5b**), KH1060 (**15**), and carborane VD (**16**) form stable interactions among the three Leu residues and the ligand, but ADTT (**3b**), TEI-9647 (**17**), and ZK168281 (**18**) do not (Figure 7). Our 25-adamantyl active vitamin D analogs shown in Table 1 are highly potent and have a variety of activity profiles. We would expect further studies of these compounds.

## 4. Materials and Methods

### 4.1. Chemistry

We carried out all reactions under an argon atmosphere. We distilled dimethoxymethane from Na/benzophenone. Reaction temperatures refer to external bath temperatures. We recorded UV spectra on a Hewlett–Packard spectrophotometer (model 8452A). We recorded the ^1^H NMR spectra in a CDCl_3_ solution on a Bruker 600 MHz spectrometer. Chemical shifts are reported in the δ scale (ppm) downfield from tetramethylsilane (δ = 0.0 ppm), using the residual solvent signal at δ = 7.26 ppm (^1^H, CDCl_3_) as the internal standard. We performed high-resolution mass spectrometric analysis (HRMS) via the atmospheric-pressure chemical ionization (APCI) method on a JMS-T100LP AccuTOF^TM^ LC-Express. We conducted high-pressure liquid chromatography (HPLC) by using Jasco PU-980 intelligent pumps equipped with an 801-SC solvent programmer and a Jasco UV-970 detector, along with a YMC Pack ODS-AM column (20 × 150 mm, particle size 5 mm, pore size 12 nm).

**(25R)- and (25S)-25-(1-Adamantyl)-1α,25-dihydroxy-2-methylidene-23-yne-19,27-dinorvitamin D_3_ (8a and 8b)** A solution of MeLi in dimethoxyethane (6 mL, 18 mmol) was added at 0 °C to an anhydrous THF solution (60 mL) of 25-ketone **11** (4.4 mg, 5.9 mmol); the mixture was stirred at that temperature for 30 min and then left at room temperature for 2 h. A saturated NH_4_Cl solution was added to the reaction, and the mixture was extracted with ethyl acetate, after which the extract was washed with saturated NaCl solution, dried over MgSO_4_, and the solvent was evaporated. The residue was chromatographed on a SiO_2_ column to yield 25-alkylated products at a ratio of approximately 1:1 for the 25-epimer pair (**12a** and **12b**) (4.0 mg, 90% yield). The mixture of 25-methylated products (**12a** and **12b**; 2.8 mg, 3.7 mmol) was dissolved in methanol (2.8 mL) and treated at 0 °C with camphorsulfonic acid (CSA) (5.1 mg, 22 mmol in methanol 400 mL) for 8 h. A saturated aqueous NaHCO_3_ solution was added to the reaction, and the mixture was extracted with ethyl acetate. The extract was washed with saline, dried over MgSO_4_, and evaporated. The residue was chromatographed on Sephadex LH-20 (1.5 g) and eluted with CHCl_3_/hexane/methanol (70/30/1) to give **8a** and **8b** as a mixture (1:1) (1.0 mg, 50.1% yield). The 25-epimer pair (**8a** and **8b**) was separated via HPLC using H_2_O/MeOH 15/85 (8.0 mL/min) as an eluent to give rapidly eluting **8a** (45.25 min) and slowly eluting **8b** (48.66 min). **8a**: ^1^H NMR (CDCl_3_) d 0.57 (3 H, s, 18-CH_3_), 1.10 (3 H, d, *J* = 6.7 Hz, 21-CH_3_), 2.58 (1 H, dd, *J* = 13.5 and 4 Hz, 4α-H), 2.82 (1 H, dd, *J* = 13.5 and 4 Hz, 9β-H), 2.85 (1 H, dd, *J* = 13.5 and 5 Hz, 10β-H), 4.50 (1 H, dd, *J* = 11.0 and 5 Hz, 1β-H or 3α-H), 4.47 (1 H, m, 3α-H or 1β-H), 5.10 and 5.11 (each 1 H, s, C(2)=CH_2_), 5.89 (1 H, d, *J* = 11.0 Hz,7-H), 6.36 (1 H, d, *J* = 11.0 Hz, 6-H). UV λmax: 244, 253, 262 nm; HRMS calcd for ^12^C_36_^1^H_51_^16^O_2_ (MH^+^-H_2_O) 515.38890, found 515.38966. **8b**: ^1^H NMR (CDCl_3_) δ 0.57 (3 H, s, 18-CH_3_), 1.09 (3 H, d, *J* = 6.5 Hz), 2.58 (1 H, dd, *J* = 13.0 and 3.5 Hz, 4α-H), 2.82 (1 H, dd, *J* = 13.5 and 4 Hz, 9β-H), 2.85 (1 H, dd, *J* = 13 and 4.5 Hz, 10β-H), 4.50 (1 H, dd, *J* = 10.0 and 5 Hz, 1β-H or 3α-H), 4.47 (1 H, m, 3α-H or 1β-H), 5.10 and 5.11 (each 1 H, s, C(2)=CH_2_), 5.89 (1 H, d, *J* = 11.0 Hz, 7-H), 6.36 (1 H, d, *J* = 11.0 Hz, 6-H). UV λmax: 244, 253, 262 nm; HRMS calcd for ^12^C_36_^1^H_51_^16^O_2_ (MH^+^-H_2_O) 515.38890, found 515.38757.

**(25R)- and (25S)-25-(1-Adamantyl)-1a,25-dihydroxy-26-methyl-2-methylidene-23-yne-19,27-dinorvitamin D_3_ (9a and 9b)** Next, the 25-ethylated analogs (**9a** and **9b**) were similarly synthesized from 25-ketone **11** via alkylation with EtLi in THF, followed by deprotection with CSA in methanol to give an isomeric mixture of **9a** and **9b,** which was separated via HPLC under the same conditions as above, with 10% water in methanol as the eluent, to give the rapidly eluting **9a** (3.36 min) and the slowly eluting **9b** (55.70 min). **9a**: ^1^H NMR (CDCl_3_) δ 0.57 (3 H, s, 18-CH_3_), 1.09 (3 H, d, *J* = 6.2 Hz, 21-CH_3_), 2.57 (1 H, dd, *J* = 13 and 3.5 Hz, 4α-H), 2.82 (1 H, dd, *J* = 13.5 and 4 Hz, 9β-H), 2.84 (1 H, dd, *J* = 13 and 4.5 Hz, 10β-H), 4.46 and 4.49 (each 1 H, each s, 1β- and 3α-H), 5.09 and 5.11 (each 1 H, each s, C(2)=H_2_), 5.89 (1 H, d, *J* = 11.0 Hz,7-H), 6.35 (1 H, d, *J* = 11.0 Hz, 6-H). UV λmax: 244, 253, 262 nm; HRMS calcd for ^12^C_37_^1^H_53_^16^O_2_ (MH^+^-H_2_O) 529.40455, found 529.40421. **9b**: ^1^H NMR (CDCl_3_) δ 0.57 (3 H, s, 18-CH_3_), 1.10 (3 H, d, *J* = 6.1 Hz, 21-CH_3_), 2.58 (1 H, dd, *J* = 13.0 and 4 Hz, 4α-H), 2.82 (1 H, dd, *J* = 12 and 3.5 Hz, 9β-H), 2.85 (1 H, dd, *J* = 13.0 and 4.5 Hz, 10β-H), 4.47 (1 H, m, 3α-H), 4.50 (1 H, d, J =10.0 and 5 Hz, 1β-H), 5.10 and 5.11 (each 1 H, s, C(2)=CH_2_), 5.89 (1 H, d, *J* = 11.0 Hz,7-H), 6.36 (1 H, d, *J* = 11.0 Hz, 6-H). UV λmax: 244, 253, 262 nm; HRMS calcd for ^12^C_37_^1^H_53_^16^O_2_ (MH^+^-H_2_O) 529.40455, found 529.40417.

**(25R)- and (25S)-25-(1-Adamantyl)-1α,25-dihydroxy-2-methylidene-26-propyl-23-yne-19,27-dinorvitamin D_3_ (10a and 10b)** The 25-adamantyl-25-butyl analogs (**10a** and **10b**) were similarly synthesized from the precursor 25-ketone (**11**) via the reaction with nBuLi, followed by the deprotection of the three hydroxy groups. The epimeric mixture of **10a** and **10b** was separated via HPLC as described above, using 9% water/MeOH as the eluent, to give faster-eluting **10a** (81.75 min) and slower-eluting **10b** (83.33 min) in a 1:1 ratio. **10a**: ^1^H NMR (CDCl_3_) δ 0.57 (3 H, s, 18-CH_3_), 1.10 (3 H, d, *J* = 6.5 Hz, 21-CH_3_), 2.58 (1 H, dd, *J* = 13.0 and 4 Hz, 4α-H), 2.82 (1 H, dd, *J* = 12 and 3.5 Hz, 9β-H), 2.85 (1 H, dd, *J* = 13.0 and 4.5 Hz, 4β-H), 4.50 (1 H, dd, *J* = 10.0 and 5.0 Hz, 1β-H or 3α-H), 4.47 (1 H, m, 3α-H or 1β-H), 5.10 and 5.11 (each 1 H, s, C(2)=CH_2_), 5.89 (1 H, d, *J* = 11.0 Hz,7-H), 6.36 (1 H, d, *J* = 11.0 Hz, 6-H). UV λmax: 244, 252.6, 262.2 nm; HRMS calcd for ^12^C_39_^1^H_57_^16^O_2_ (MH^+^-H_2_O) 557.43585, found 557.43400. **10b**: ^1^H NMR (CDCl_3_) δ 0.57 (3 H, s, 18-CH_3_), 1.10 (3 H, d, *J* = 6.5 Hz, 21-CH_3_), 2.57 (1 H, dd, *J* = 13.0 and 4.0 Hz, 4α-H), 2.82 (1 H, dd, *J* = 12.0 and 3.5 Hz, 9β-H), 2.85 (1 H, dd, *J* = 13.0 and 4.5 Hz, 10β-H), 4.50 (1 H, dd, *J* = 10.0 and 5.0 Hz, 1β-H or 3α-H), 4.47 (1 H, m, 3α-H or 1β-H), 5.10 and 5.11 (each 1 H, s, C(2)=CH_2_), 5.89 (1 H, d, *J* = 11.0 Hz,7-H), 6.36 (1 H, d, *J* = 11.0 Hz, 6-H). UV λmax: 244, 253, 262 nm; HRMS calcd for ^12^C_39_^1^H_57_^16^O_2_ (MH^+^-H_2_O) 557.43585, found 557.43100.

### 4.2. Plasmids

We used the expression plasmids pCMX-VDR, pCMX-VP16- VDR, pCMX-GAL4-RXRα, pCMX-GAL4-SRC-1, and pCMX-GAL4- N-CoR, as reported previously [43]. In pCMX-GAL4-SRC-1 and pCMX-GAL4-NCoR, the nuclear receptor-interacting domains of SRC-1 (amino acids 595–771; GenBank accession code U90661) and NCoR (amino acids 1990–2416; GenBank accession code U35312) were inserted into the pCMXGAL4 plasmid, respectively. We used luciferase reporters: VDR-responsive Sppx3-tk-LUC and GAL4-responsive MH100(UAS) × 4-tk-LUC reporters [43]. We also used the pGEX-VDR plasmid, in which VDR-LBD (amino acids 140–427) was inserted into the pGEX vector (GE Healthcare, Chicago, Il, U.S.A.) to generate glutathione transferase (GST) fusion proteins [56].

### 4.3. Vitamin D Receptor-Binding Assay

The pGEX-hVDR and pGEX were expressed for a GST-VDR fusion protein and a control GST protein, respectively, in *Escherichia coli* BL21 (Merck, Darmstadt, Germany). The bacteria were lysed by sonication in a sonication buffer (50 mM Tris-HCl, pH 8.0, 50 mM NaCl, 1 mM EDTA, and 1 mM DTT). The supernatant proteins (1 μg) were diluted in a binding buffer (25 mM Tris, pH 7.5, 100 mM KCl, 25 mM DTT, 4 mM CHAPS, pH 7.5) containing bovine serum albumin (100 μg/mL). A solution containing each test compound in 15 μL of EtOH was added to 570 μL of the GST protein solution in each tube. After being vortexed 2–3 times, the mixture was incubated for 30 min at room temperature. Then, [26,27-methyl-^3^H]-1,25(OH)_2_D_3_ (PerkinElmer, Waltham, MA, USA) in 15 μL of ethanol was added. After being vortexed 2–3 times, the whole mixture was allowed to stand at 4 °C for 20 h, and 400 μL of dextran-coated charcoal solution (Sigma) was added to remove free ligands. After 30 min at room temperature, bound and free [^3^H]-1,25(OH)_2_D_3_ were separated via centrifugation at 3000 rpm for 10 min at 0 °C, and aliquots (800 μL) of the supernatant were mixed with 9.2 mL of Bio Fluor (PerkinElmer) and submitted for radioactivity counting.

### 4.4. Cell Line Cultures

HEK293 human kidney cells (RIKEN Cell Bank, Tsukuba, Japan) were cultured in Dulbecco’s modified Eagle’s medium (DMEM) (Fujifilm Wako Pure Chemical Corporation, Tokyo, Japan) containing 5% fetal bovine serum (FBS), 100 U/mL penicillin, and 0.1 mg/mL streptomycin (Nacalai Tesque, Kyoto, Japan). SW480 human colon carcinoma cells (American Type Culture Collection, Manassas, VA) and HaCaT immortalized keratinocyte cells (kindly provided by Dr. Tadashi Terui, Department of Dermatology, Nihon University School of Medicine) were cultured in DMEM containing 10% FBS, 100 U/mL penicillin, and 0.1 mg/mL streptomycin. MG63 human osteosarcoma cells (RIKEN Cell Bank) and lung-derived H292 cells (American Type Culture Collection) were cultured in minimum essential medium containing 10% FBS, 100 U/mL penicillin, and 0.1 mg/mL streptomycin. U937 human myeloid leukemia cells (RIKEN Cell Bank) were cultured in RPMI 1640 medium (Fujifilm Wako) containing 10% FBS, 100 U/mL penicillin, and 0.1 mg/mL streptomycin. All cells were cultured at 37 °C in a humidified atmosphere containing 5% CO_2_.

### 4.5. Luciferase Reporter Assays for VDR Transactivation and Mammalian Two-Hybrid Assays

Transfections of 50 ng of TK-Spp × 3-LUC reporter plasmid, 10 ng of pCMX-β-galactosidase, and 15 ng of pCMX-VDR for each well of a 96-well plate were performed in HEK293 cells using the calcium phosphate coprecipitation method, as described previously [43]. The mammalian two-hybrid assay for cofactor interaction with VDR used 50 ng of TK-MH100(UAS) × 3- LUC reporter plasmid, 10 ng of pCMX-β-galactosidase, 15 ng of pCMX-GAL4-RXRα, pCMX-GAL4-SRC1, or pCMX-GAL4-NCoR, and 15 ng of pCMX-VP16-VDR for each well of a 96-well plate. Luciferase and β-galactosidase activities were measured with a luminometer and a microplate reader (Molecular Devices, Sun Jose, CA, USA), respectively. Luciferase values were normalized to the internal β-galactosidase control.

### 4.6. Reverse Transcription and Quantitative Real-Time PCR Analysis

For gene expression analysis, 1 × 10^4^ cells per well were plated in a 24-well plate. After 24 h, the cells were treated with an ethanol control or 100 nM of each test compound for 24 h. The total RNAs from the samples were extracted using the acid guanidine thiocyanate phenol/chloroform method [57], and cDNAs were synthesized using the ImProm-II reverse transcription system (Promega, Madison, WI) [43]. Real-time PCR was performed on the ABI PRISM 7000 sequence detection system (Thermo Fisher Scientific, Waltham, MA, USA) using Power SYBR Green PCR master mix (Thermo Fisher Scientific). The primer sequences were as follows: CYP24A1 5′-TGAACGTTGGCTTCAGGAGAA-3′ and 5′-AGGGTGCCTGAGTGTAGCATCT-3′; GAPDH 5′-ACTTCGCTCAGACACCATGG-3′ and 5′-GTAGTTGAGGTCAATGAAGGG-3′. The mRNA values were normalized to the mRNA levels of GAPDH and calculated relative to those in the 1,25(OH)_2_D_3_ treatment.

### 4.7. Human Dedifferentiated Fat Cell Isolation and Culture

Samples of human subcutaneous adipose tissue were obtained from patients who underwent surgery in the Department of Pediatric Surgery at Nihon University Itabashi Hospital (Tokyo, Japan). The patients gave their written informed consent, and the Ethics Committee of Nihon University School of Medicine approved this study. We prepared dedifferentiated fat (DFAT) cells using the ceiling culture method, as described previously [50]. Briefly, the adipose tissue (approximately 1 g) was cut into small pieces and digested with a 0.1% type I collagenase solution (Koken Co., Ltd., Tokyo, Japan). Cell samples were filtered and centrifuged at 135× g for 3 min, followed by collection of the floating cell layer containing mature adipocytes. The isolated adipocytes were washed with phosphate-buffered saline (PBS) and plated in 25 cm^2^ culture flasks (Thermo Fisher Scientific) filled completely with CSTI-303MSC (Cell Science and Technology Institute, Miyagi, Japan) containing 20% FBS with 5 × 10^4^ cells per flask. The cells that adhered to the upper surface of the flasks were cultured for conversion to dedifferentiated fat cells. Fibroblast-like adhered cells (dedifferentiated fat cells) were cultured in 5 mL of CSTI-303MSC containing 20% FBS. Cells at passages 2–4 were used for differentiation experiments.

### 4.8. Osteogenic Differentiation Assay

Cells were grown to confluency in 24-well plates and incubated for a week in DMEM containing 10% FBS, 100 nM dexamethasone (Merk), 10 mM β-glycerophosphate (Merk), and 50 mM L-ascorbic acid-2-phosphate (Merk), with 100 nM of each compound. The induction medium was replaced on day 4. The alkaline phosphatase (ALP) activity in the cell lysates was determined on day 7 with a lab assay ALP kit (Fujifilm wako). The total protein content was determined with the BCA protein assay kit (Thermo Fisher Scientific). ALP levels were normalized to the total protein content.

### 4.9. Statistical Analysis

Data are presented as means ± S.D. We performed one-way ANOVA followed by Tukey’s multiple comparisons to assess significant differences using Prism 8 (GraphPad Software, La Jolla, CA, USA).

### 4.10. Modeling and Computation

We performed first-principals (quantum mechanical: QM) calculations for the ligand-bound VDR-LBD complexes based on the ab initio fragment molecular orbital (FMO) method [58,59] to quantitatively evaluate the interaction energies between various ligands and the VDR-LBD. The natural ligand 1,25(OH)_2_D_3_ (**1**), partial agonists ADTK1 (**5b**), ADKM2 (**8b**), ADKM4 (**9b**), and ADKM6 (**10b**), and antagonist ADTT (**3b**) were adopted for the QM computational targets.

We refined and rebuilt the atomic coordinates of the crystal structures of the hVDR-LBD complex with natural ligand **1**. The protocols [60] of our refinement and rebuilding of the crystal structures of the protein–ligand complex were as follows: (i) the refinement of the target complex structures was optimized using the PDB_REDO web server [61]. This server optimizes various refinement parameters (including B-factor weight, X-ray weight, TLS groups, and bulk solvent modeling), chooses between an anisotropic or isotropic B-factor model, rebuilds side chains in rotamer conformations, flips side-chains to optimize hydrogen bond networks, checks peptides for ‘flipping’, re-evaluates the water model, and validates all of the present ligands. The R and R-free values of the complexes improved from 0.1910 and 0.2140 to 0.1501 and 0.1826, respectively. (ii) The missing hydrogen atoms were added by using the Protonate 3D module [62] within the Molecular Operating Environment (MOE) program [63] under the standard conditions (pH = 7.0, 25 °C). The orientations of the added hydrogen atoms were then optimized by using an energy minimization scheme through molecular mechanics (MMs) calculations utilizing an Amber10 EHT force field under the generalized Born solvation, which uses Amber10 parameters for macromolecules and extended Hückel theory parameterization for small molecules, which takes electronic effects into account and is incorporated in the MOE program [63].

The positions of important hydrogens in the hydrogen network associated with the three OH groups in ligand **1** were determined as in our previous procedure [64,65] and were optimized using QM calculations at the standard B3LYP/6-31G** level using the Gaussian09 program [66] for the sub-model system of the corresponding six amino acid residues (Ser237, Arg274, Tyr143, Ser278, His305, and His397) and ligand **1**.

A similar modeling process was applied to the complex structures of rVDR-LBD with ADTK1 (**5b**) (PDB ID: 3VTB) and rVDR-LBD with ADTT (**3b**) (2ZMI) for the FMO calculation.

Since the structures of the VDR-LBD complex bound by ADKM2 (**8b**), ADKM4 (**9b**), and ADKM6 (**10b**) have not been experimentally determined, computational modeling structures of ADKM2 (**8b**), ADKM4 (**9b**), and ADKM6 (**10b**) were constructed as follows:

Firstly, the model structures of ADKM2 (**8b**), ADKM4 (**9b**), and ADKM6 (**10b**) were constructed by replacing a hydrogen with a methyl, ethyl, and n-Bu group, respectively. The model structure of the ligand was computationally determined in an isolated system via partial optimization for only the position 25 substituent, in which the absolute conformation at position 25 was maintained at the B3LYP/6-31G** level using the Gaussian09 program [66].

Secondly, the modeled structures of the ADKM2 (**8b**)-, ADKM4 (**9b**)-, and ADKM6 (**10b**)-bound rVDR-LBD complexes were computationally constructed by superimposing ADKM2 (**8b**), ADKM4 (**9b**), and ADKM6 (**10b**) onto a bound ADTK1 **5b** already in the LBP of rVDR-LBD (3VTB) and then deleting the bound ADTK1 **5b**.

In this study, all FMO calculations for the ligand-bound VDR-LBD complexes were carried out with the correlated resolution-of-identity (RI)-MP2 method [67,68] subjected to counterpoise corrections with the correlation-consistent double ζ(cc-pVDZ) basis set [69] level, using the PAICS program package [70].

The correlated FMO calculation can estimate the interaction energy quantitatively, including not only electrostatic interactions but also London dispersion forces (van der Waals interactions), such as CH/π–π stacking. Utilizing this calculation, FMO-IFIEs were obtained for the various ligand-bound VRD-LBD complexes. The correlated FMO-IFIEs at the MP2 level are represented as the sum of the HF-IFIEs (mainly electrostatic energies) and the ΔMP2-IFIEs (mainly dispersion force) (see the Appendix A).

The ΔMP2-IFIE was adopted to evaluate the stable/unstable interaction between a hydrophilic and/or hydrophobic amino acid residue and the ligand in the VDR-LBD complex.

The root-mean-square deviations (RMSDs) of all of the atoms/Cα of each residue pair were calculated by using the plugin script RmsdByResidue [71] of the molecular graphics software package PyMOL [72].

## Data Availability

Not applicable.

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
