# Peer review of "Syntheses of 25-Adamantyl-25-alkyl-2-methylidene-1α,25-dihydroxyvitamin D3 Derivatives with Structure–Function Studies of Antagonistic and Agonistic Active Vitamin D Analogs"

_biomolecules, 2023, doi:10.3390/biom13071082_

Round 1

Reviewer 2 Report

The presented article describes preparation of three novel vitamin D derivatives substituted at C-25 with adamantyl group. It is surprising that the presence of such a bulky group located at the terminus of the steroidal side chain does not prevent the analogs from binding to the nuclear receptor (VDR). A synthetic value of this article is not high because the described synthesis of the final compounds consists of the two relatively simple steps. However, the prepared vitamins seem to be interesting targets. Taking into account their interesting biological properties, the continuation of the studies on the related vitamin D compounds has been undoubtedly justified.

Moreover, the value of the reviewed article has been significantly enhanced by detailed biological studies of the synthesized compounds (VDR affinity and transactivation activity, evaluation of their effects on the VDR interactions and endogenous gene expression, etc). The Authors calculated also model structures of the VDR complexes with the synthesized analogs (and some model compounds) and discussed structure-activity relationship of the agonistic and antagonistic vitamins.

Taking all these facts into consideration this Reviewer recommends the article for publication in Biomolecules if the Authors revise the manuscript taking into account the comments shown in the attached file.

Shown in the attached file

Reviewer 3 Report

The authors designed and synthesized novel vitamin D derivatives with an alkyl group substituted at C-25 of the previously obtained side-chain adamantane substituted analogs aiming to develop even more cell-selective VDR ligands. The study allowed for a valuable conclusion that bulky substituents introduced in the vitamin D side-chain at C-24, C-25, or C-26  increased stability and tissue selectivity. 

The study is very well-designed, conducted carefully, and described precisely. I do not have any major comments. 

Minor comments:

- the Results and Discussion part is very interesting, but it is quite extensive. In such a case, adding a short Conclusions section, as suggested in the "Instructions for the authors",  might be convenient for the readers to get a quick inside into the major achievements of the study

- as the absolute stereochemistry at C-25 was determined indirectly, by comparisons of biological activities, the 25S and 25R stereochemistry should be reported throughout the manuscript as "tentative"

- line 128: in the shortened chemical name of analogs the 2-methylidene  substituent should also be indicated as important for activity

- line 130: "11" should be in bold, as for compounds, and 11 is certainly a 25- and not 24-ketone, as indicated in line 446

- line 465, 470, 486, and 503,  - lambda max and not lmax

Minor editing of the English language might be suggested. 

Round 2

Reviewer 2 Report

The manuscript has been significantly improved. However, in my opinion, a weird nomenclature (didehydro, tetradehydro) of the synthesized compounds 8a,b-10a,b should be definitely changed. Moreover, the Authors do not seem to notice that methyl carbons 26 and 27 belong to the vitamin D3 carbon skeleton. Therefore, 8a is not “25-methyl” derivative because this methyl belongs to the steroidal skeleton. Consequently, 9a is not “25-ethyl” derivative but 26-methyl derivative, etc. I strongly suggest that the following names should be used in the experimental part and throughout the text:

(25R)-(1-Adamantyl)-1a,25-dihydroxy-2-methylidene-23-yne-19,27-dinorvitamin D3 (8a)

(25S)-(1-Adamantyl)-1a,25-dihydroxy-2-methylidene-23-yne-19,27-dinorvitamin D3 (8b)

(25R)-(1-Adamantyl)-1a,25-dihydroxy-26-methyl-2-methylidene-23-yne-19,27-dinorvitamin D3 (9a)

(25S)-(1-Adamantyl)-1a,25-dihydroxy-26-methyl-2-methylidene-23-yne-19,27-dinorvitamin D3 (9b)

(25R)-(1-Adamantyl)-1a,25-dihydroxy-2-methylidene-26-propyl-23-yne-19,27-dinorvitamin D3 (10a)

(25S)-(1-Adamantyl)-1a,25-dihydroxy-2-methylidene-26-propyl-23-yne-19,27-dinorvitamin D3 (10b)
